# Spatiotemporal Vegetation Variability and Linkage with Snow-Hydroclimatic Factors in Western Himalaya Using Remote Sensing and Google Earth Engine (GEE)

Dhiraj Kumar Singh [1], Kamal Kant Singh [2], George P. Petropoulos [3], Priestly Shan Boaz [1], Prince Jain [4], Sartajvir Singh [5], Dileep Kumar Gupta [6,*] and Vishakha Sood [7]

1   Kalpana Chawla Centre for Research in Space Science & Technology (KCCRST), Chandigarh University, Mohali 140413, India; dhiraj.e11719@cumail.in (D.K.S.); kccrsst@cumail.in (P.S.B.)
2   Defence Geoinformatics Research Establishment (DGRE), Him-Parisar, Sector-37A, Chandigarh 160036, India; kamal.kant.dgre@gov.in
3   Department of Geography, Harokopio University of Athens, El. Venizelou St., 70, 17671 Athens, Greece; gpetropoulos@hua.gr
4   Department of Mechatronics Engineering, Parul University, Vadodara 391760, India; prince.jain24312@paruluniversity.ac.in
5   University Institute of Engineering, Chandigarh University, Mohali 140413, India; Sartajvir.dhillon@ieee.org
6   Department of Physics, Banaras Hindu University, Varanasi 221005, India
7   Department of Civil Engineering, Indian Institute of Technology, Ropar 140001, India; vishakha.sood@ieee.org
*   Correspondence: dileepgupta85@gmail.com

**Abstract:** The mountain systems of the Himalayan regions are changing rapidly due to climatic change at a local and global scale. The Indian Western Himalaya ecosystem (between the tree line and the snow line) is an underappreciated component. Yet, knowledge of vegetation distribution, rates of change, and vegetation interactions with snow-hydroclimatic elements is lacking. The purpose of this study is to investigate the linkage between the spatiotemporal variability of vegetation (i.e., greenness and forest) and related snow-hydroclimatic parameters (i.e., snow cover, land surface temperature, Tropical Rainfall Measuring Mission (TRMM), and Evapotranspiration (ET)) in Himachal Pradesh (HP) Basins (i.e., Beas, Chandra, and Bhaga). Spatiotemporal variability in forest and grassland has been estimated from MODIS land cover product (MCD12Q1) using Google Earth Engine (GEE) for the last 19 years (2001–2019). A significant inter- and intra-annual variation in the forest, grassland, and snow-hydroclimatic factors have been observed during the data period in HP basins (i.e., Beas, Chandra, and Bhaga basin). The analysis demonstrates a significant decrease in the forest cover (214 ha/yr.) at the Beas basin; however, a significant increase in grassland cover is noted at the Beas basin (459 ha/yr.), Chandra (176.9 ha/yr.), and Bhaga basin (9.1 ha/yr.) during the data period. Spatiotemporal forest cover loss and gain in the Beas basin have been observed at ~7504 ha (6.6%) and 1819 ha (1.6%), respectively, from 2001 to 2019. However, loss and gain in grassland cover were observed in 3297 ha (2.9%) and 10,688 ha (9.4%) in the Beas basin, 1453 ha (0.59%) and 3941 ha (1.6%) in the Chandra basin, and 1185 ha (0.92%) and 773 ha (0.60%) in the Bhaga basin, respectively. Further, a strong negative correlation (r = −0.65) has been observed between forest cover and evapotranspiration (ET). However, a strong positive correlation (r = 0.99) has been recorded between grassland cover and ET as compared to other factors. The main outcome of this study in terms of spatiotemporal loss and gain in forest and grassland shows that in the Bhaga basin, very little gain and loss have been observed as compared to the Chandra and Beas basins. The present study findings may provide important aid in the protection and advancement of the knowledge gap of the natural environment and the management of water resources in the HP Basin and other high-mountain regions of the Himalayas. For the first time, this study provides a thorough examination of the spatiotemporal variability of forest and grassland and their interactions with snow-hydroclimatic factors using GEE for Western Himalaya.

**Keywords:** vegetation; remote sensing; Western Himalaya; snow-hydroclimatic factors; Google Earth Engine

## 1. Introduction

Vegetation is a vital factor in water–soil–atmosphere systems that controls the global energy exchange and hydrological, biogeochemical, and carbon cycles [1,2]. Additionally, it controls the climatic and biological balance of the planet and offers resistance to erosion and sediment movement [3]. Therefore, using a variety of monitoring indices, vegetation may operate as an excellent indicator for identifying and monitoring variations in environmental, meteorological, and hydrological elements such as precipitation, temperature, evapotranspiration, and soil moisture.

About 3.9 billion hectares (ha) of the world is covered by forest, which is ~31% of the total land area. Since 1990, 178 million ha of forest area have disappeared from the world, an area almost equal to Libya (https://www.fao.org/3/CA8753EN/CA8753EN.pdf (accessed on 5 February 2023). The rate of forest loss reduced significantly from 1990 to 2020 because of decreased deforestation in certain nations and increased forest area in other nations. The net forest loss across the globe was reported to be decreasing at the rate of 7.8, 5.2, and 4.7 million ha per year during 1990–2000, 2000–2010, and 2010–2020, respectively (Global Forest Resources Assessment, 2000 https://www.fao.org/forestry/fra/86624/en/, accessed on 12 February 2023). In 2013, India had a forest cover of 21.17%, with variations in percentage seen between the Indian Himalayan Region (IHR) and other regions within the nation. India's forest area comprises 75.52% of the total land area. The Indian Himalayan Region is included in the country's states; however, in terms of India's geographical expanse, it accounts for a mere 15.7%. The Forest Policy of India in 1988 was formulated with the objective of preserving and sustaining 66% of the forest cover in the country. The study conducted by Negi [4] focused on a specific geographical region characterized by hilly terrain and covered by a dense forest. Further, according to the Forest Survey of India (FSI), the total forest area in India was about 76.5 million ha in 2001 (https://frienvis.nic.in/Database/Forest-Cover-in-India-2001-05_2248.aspx, accessed on 12 February 2023). However, changes in forest and vegetation are dynamic processes, and the vegetation area covered changes yearly.

During the last five decades, various approaches for vegetation estimation have been adopted, i.e., field-based measurement, remote sensing data, and GIS-based methods. Field-based measurement methods are the most accurate ways to collect vegetation information. However, they have limitations in providing spatial distribution of vegetation cover over a large area [5]. Satellite-based remote sensing images with high relationships between spectral bands and vegetation parameters have the potential to estimate vegetation area for a large geographical region with a high spatial and temporal resolution, especially in rugged inaccessible terrain [6,7].

Remote sensing satellite images-derived vegetation indices calculate vegetation greenness to estimate leaf chlorophyll content, leaf area, greenness, and forest cover area [8–12]. Vegetation indices in the past have been used to estimate vegetation cover on a local to global scale [13] using a wide variety of remote sensing satellite products (e.g., Landsat series, Moderate Resolution Imaging Spectroradiometer (MODIS), Advanced Wide Field Sensor (AWiFS), Advanced Spaceborne Thermal Emission and Reflection Radiometer (ASTER), WorldView-1, WorldView-2, QuickBird, GeoEye, IKONOS, and SPOT, etc.). The linkage of vegetation biophysical features to red/infrared and other spectral index ratios utilizing reflectance characteristics observed in multispectral bands was developed in 1974 by Rouse et al. [14]. After the Landsat satellite sensor launch, various mathematical spectral bands combinations vegetation indices have been established. Rouse et al. [14] developed a Normalized Difference Vegetation Index (NDVI) to discriminate vegetation from other land features. Another similar vegetation index, the Difference Vegetation Index (DVI), used to distinguish the soil and vegetation for small vegetation-covered areas, was established by Jackson et al. [15]. Further, Kim et al. [16] proposed a Chlorophyll Absorption Ratio Index (CARI) to estimate the amount of chlorophyll absorbed in vegetation. The Renormalized Difference Vegetation Index (RDVI) was established to overcome the limitations of DVI and NDVI [17]. RDVI integrates the potential of DVI and NDVI and is further used to

estimate the global photosynthetically active radiation absorbed by vegetation. Further, a Modified Chlorophyll Absorption Ratio Index (MCARI) was developed to estimate the leaf chlorophyll concentration in the plant [18]. A Triangular Vegetation Index (TVI) was established to describe the energy absorbed as the function of the difference in red and NIR reflectance bands [19]. The Three-band Gradient Difference Vegetation Index (TGDVI) was introduced to overcome the limitation of the lower saturation point of NDVI [20]. According to published studies such as Glenn et al. [21], vegetation indices and canopy photosynthesis have a linear relation. Typically, vegetation indices are calculated based on the link between the robust near-infrared reflectance of the chlorophyll in plants and the absorption of visible light. According to the above studies, it can be inferred that the use of multiple vegetation indices is essential for improving the predictions of vegetation patterns and dynamics because seasonal and interannual vegetation patterns cannot be determined solely by relying on a single index due to its various limitations. However, accurate vegetation monitoring using the above index is difficult because it is region-specific (land cover type). Therefore, land cover products have been generated for the entire globe with good accuracy.

Numerous studies have connected various hydroclimatic parameters with spatiotemporal fluctuations of vegetation across the globe [22–26], including very few for Indian Western Himalaya. However, the linkage with the snow-hydroclimatic factor is still lacking. The fast global climate change in recent years has drawn attention to connections between vegetation and climate [1]. However, predominantly due to a large variability at the regional scale in terms of vegetation types and climate, there has not been much study undertaken in India to show how plant phenology and snow-hydroclimatic parameters relate to one another, particularly in Himalayan hilly regions. This needs more attention, as the Indian Himalayan areas offer several classes of flora, soil, temperature, and height, ranging from the summits of high mountains to tropical woodlands in the lowlands [27].

In recent years, the vegetation pattern has significantly affected eco-hydro-geomorphic features in the Himalayan region, which supplies water to many downstream areas [28]. Nouri et al. [29] studied vegetation estimation using various vegetation indices and observed NDVI as a robust indicator to study vegetation characteristics. Spatiotemporal vegetation information will significantly contribute to the area of environmental management by assisting stakeholders and policymakers in identifying the early warning signals of deterioration or improvement in land conditions in any particular region. Understanding the link between snow-hydroclimatic parameters and vegetation in the Himalayan area has not received much attention.

In view of the above, in this study, a long-term spatiotemporal pattern of the forest and grassland cover is investigated at three basins (Beas, Chandra, and Bhaga basins), which are located in the Indian state of Himachal Pradesh. Even with the advancement in geospatial technology for monitoring vegetation using Google Earth Engine (GEE) it has not been thoroughly explored in the literature until the present time. In this study, the relationship between the spatiotemporal variations of forest and grassland cover and the linkage with snow-hydroclimatic factors in the study areas has been examined. MODIS yearly Land-Cover Product (i.e., MOD12Q1) for the past 19 years (2001–2019) have been used to estimate the forest and grassland cover using GEE in the study area, and cloud-free snow cover, and hydrochloric factors (land surface temperature (LST), Precipitation (PPT), and Evapotranspiration (ET)) were analyzed to understand the relation between them. Additionally, spatiotemporal loss and gain in forest and grassland have been estimated in the study area during the data period. The results and analysis of this study can contribute to our understanding of the eco-hydrological features of the studied basins region of the Himalayas. This study provides, for the first time, a systematic examination of the spatiotemporal variability of vegetation (forest and grassland) and their interactions with snow-hydroclimatic factors in Western Himalaya.

## 2. Study Area

The present study focuses on three Himachal Pradesh (HP) basins (i.e., Beas, Chandra, and Bhaga basins) located in the lower Western Himalaya range, India, as shown in Figure 1. The geographical area of the Indian State, i.e., HP, is 55,673 km², which constitutes 1.69% of the total area of the nation. The HP basins are bounded by four states (Jammu and Kashmir in the north, Haryana in the south, Punjab in the west, and Uttarakhand in the southwest), the union territory of Ladakh, and the China international boundary in the east. The study area is located between 32°N and 33°N latitude and 76°54′E and 77°55′E longitude. The state's major rivers are the Yamuna, Chenab, Beas, Ravi, and Satluj.

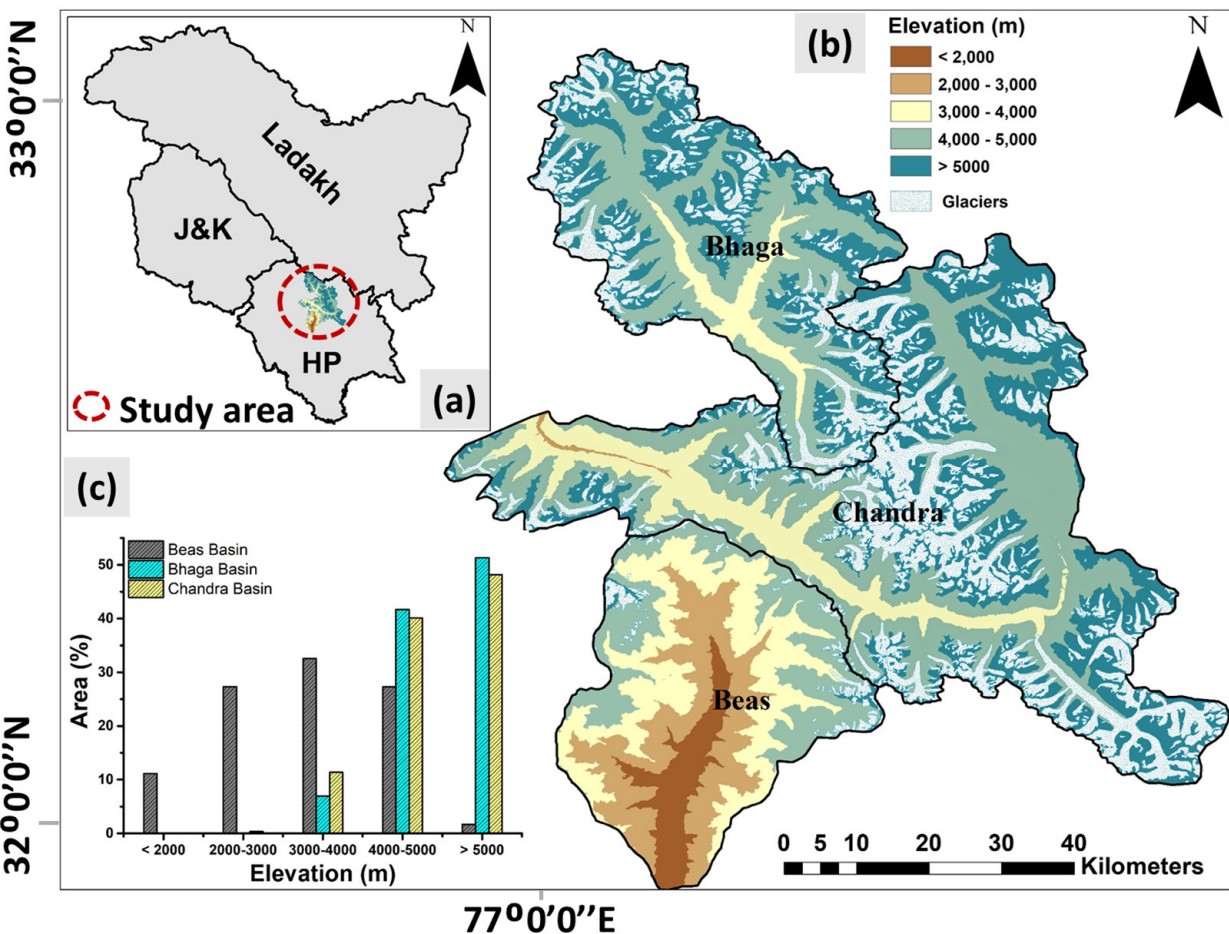

**Figure 1.** (**a**) Geographical location of the study area, (**b**) spatial elevation map of three HP basins (i.e., Beas, Chandra, and Bhaga), and (**c**) hypsometric curve of Beas, Chandra, and Bhaga basins.

The lower Himalayas, also known as the subtropical area, have an altitude between 2000 and 4000 m (a.s.l.) and are heavily blanketed with seasonal snow throughout the winter. With its frequent snowfall, moderate temperatures, and thick snow cover, it has many features of a marine snow climate. The region has a third of its land continuously covered with snow, glaciers, and a freezing desert. Due to the challenging environment, this area has little tree growth and about an average of 1353.4 mm annual rainfall has been observed by Jaswal et al. [30] from 1951 to 2015. The seasonal snow cover's peak accumulation period in this range typically lasts from mid-December to mid-March. With each snowstorm, the snow cover gets deeper, reaching its maximum depth in late February or the first week of March. However, due to an increase in ambient temperature, snow cover begins to melt starting in mid-March, and the whole snowpack melts by late April or the first week of May [31].

The mean altitude of the Beas, Chandra, and Bhaga basins are ~3310, 4848, and 4928 m (a.s.l.) and they lie between 1177 to 6017 m, 2830 to 6541 m, and 3272 to 6443 m (a.s.l.), respectively. The total area of the Beas, Chandra, and Bhaga basins is 1137 km$^2$, 2463 km$^2$, and 1288 km$^2$, respectively. The Beas basin receives heavy snowfall from December to March every year, and the average maximum, minimum, and ambient temperature were observed at ~7.9 °C, −1.3 °C, and 2.6 °C, respectively, during the winter season of November–April (1989–2017) [32]. The Bhaga and Chandra are the sub-basins of the Chenab basin located in the Lahaul–Spiti district of HP and receive snowfall in the winter period from October to April every year [33]. These basins are extensively glaciated, with the Chandra Basin covering 703.64 km$^2$ [34] and the Bhaga Basin covering 376 km$^2$ [35]. The Chandra basin consists of a valley and mountain kinds of glaciers with an average slope of 21° and aspects of north, northeast, and northwest. The basin climate is thought to be impacted by western disturbances in the winter and monsoon circulation in the summer, according to a meteorological station close to the basin. The average monthly temperature was 8.9 °C, and precipitation was 63.9 mm, with the highest precipitation of 177.6 mm/month in the month of July from 1901 to 2001 [36]. Snow-hydroclimatic data in the study area have been analyzed in four different seasons, i.e., (a) winter season (January to February), (b) pre-monsoon season (March to June), (c) monsoon season (July to September), and (d) post-monsoon season (October to December).

Figure 1c represents the hypsometry graph of the Beas, Chandra, and Bhaga basins. Beas basin area in elevation ranges below 2000 m, 2000–3000 m, 3000–4000 m, 4000–5000 m, and above 5000 m (a.s.l.) is about 11%, 27%, 33%, 27%, and 2%, of the total basin area, respectively. However, Chandra and Bhaga basins have an area of 0.4%, 11.6%, 40%, and 48%, and 0%, 7%, 42%, and 51% in the elevation range 2000–3000 m, 3000–4000 m, 4000–5000 m, and above 5000 m (a.s.l.), respectively.

## 3. Geospatial Data and Methods

### 3.1. Forest and Grassland

The International Geosphere-Biosphere Program (IGBP) MCD12Q1 Version 6 Level 3 products of the MODIS yearly land cover datasets from 2001 to 2019 have been used to obtain forest and grassland [37]. The product has been downloaded from Google Earth Engine (GEE) (https://developers.google.com/earth-engine/datasets/catalog/MODIS_006_MCD12Q1, 8 September 2022) at 500 m spatial resolution. This land cover product is generated from MODIS Terra and Aqua satellite data using supervised classifications and additional post-processing, having 75% classification accuracy [38]. The MCD12Q1 product contains 17 land cover types with 11 types of natural vegetation, three types of developed and mosaic lands, and three types of non-vegetation land [39]. In this study, MCD12Q1 was reclassified into two major classes: forest and grassland (Table 1). Further, the reclassified land cover product has been used to analyze the spatiotemporal variability of the forest and grassland in the Beas, Chandra, and Bhaga basins during the data period.

**Table 1.** MCD12Q1 land cover product classes and reclassified land type.

| IGBP Class Code | MCD12Q1 Land Type | Reclassified Land Type |
|:---:|:---|:---|
| 1 | Evergreen needleleaf forest | |
| 2 | Evergreen broadleaf forest | |
| 3 | Deciduous needleleaf forest | Forest |
| 4 | Deciduous broadleaf forest | |
| 5 | Mixed forest | |
| 6 | Closed shrublands | |
| 7 | Open shrublands | |
| 8 | Woody savannas | Grassland |
| 9 | Savannas | |
| 10 | Grasslands | |



### 3.2. Hydroclimatic Datasets

Precipitation data have been retrieved from the current version of Tropical Rainfall Measuring Mission (TRMM) at 0.25° × 0.25° spatial resolution from the GEE (https://developers. google.com/earth-engine/datasets/catalog/TRMM_3B43V7, 8 September 2022) during the data period from 2001 to 2019. Monthly rainfall data estimated from the 3B437 product provide a precipitation map by merging precipitation products and various remote sensing-based satellite data with rain gauge data [40]. Shukla et al. [41] evaluated the TRMM precipitation data over the Himalayan basin and observed a good correlation coefficient ($R^2$) with ground data, i.e., ~0.65 for post-monsoon and ~0.57 for the winter season.

MODIS LST products at 1 km × 1 km spatial resolution are available worldwide at daily, 8-day, and monthly intervals. In the paper, the monthly MODIS LST (MOD11A1) product has been downloaded from GEE (https://developers.google.com/earth-engine/datasets/catalog/MODIS_006_MOD11A1, 8 September 2022) for the study period. Many ground measurements have been used to calibrate this product, particularly in the Himalayan area [42]. Kenawy et al. [43] reported that the accuracy of this daily LST product might be affected by heavy aerosols. Therefore, in the present study, a composite monthly product image for the study area was created using GEE.

MODIS ET data are available for the entire globe at 500 m spatial resolution for every 8-day, monthly, and annual period. In this paper, annual ET data has been downloaded from GEE (https://developers.google.com/earth-engine/datasets/catalog/MODIS_006_MOD16A2, 8 September 2022) during the data period for the study area. Other studies, such as that of Srivastava et al. [2] reported that the MODIS-derived ET product has a good correlation with ground observation for Indian regions.

### 3.3. MODIS Daily Cloud-Free Snow Cover Product

In the present research, the daily cloud-free enhanced MODIS snow product (available at https://doi.org/10.1594/PANGAEA.918198, 8 September 2022) was utilized to estimate the snow cover area (SCA) for the study period. Biases in daily MODIS Terra and Aqua products owing to sensor limitations and cloud cover have been addressed in this product for high-mountain Asia using a cloud removal technique followed by a gap-filling algorithm [37]. The product (M*D10A1GL06) was created at a spatial resolution of 500 m by combining daily MODIS Terra and Aqua products, and then merging them with Randolph Glacier Inventory version 6.0 (RGI 6.0). Muhammad and Thapa [37] evaluated this product for the high mountain Asia regions and found it to be less uncertain (i.e., 32.9% for cloud cover and 6.2% for sensor restriction due to big sun zenith angle) than previous MODIS SCA products.

### 3.4. Method of Evaluation

In this study, vegetation (i.e., forest and grassland) and snow-hydroclimatic factor data were analyzed using remote sensing and GEE. The three HP basins (i.e., Beas, Chandra, and Bhaga basins) boundary file was used to extract the forest and grassland land cover from the yearly MODIS land cover product by setting the same geographical project in GEE. The extracted MODIS land cover product for the study area has 17 classes; therefore, we reclassify it into two classes (forest and grassland) in the GEE image library using the image processing and analysis toolbox.

Spatiotemporal variation in yearly forest and grassland areas was extracted in the study region for the 19-year data period (2001–2019). Similarly, snow-hydroclimatic factors pixel values and area were estimated in the study region, and average monthly and yearly data were generated over the entire 19-year period. Long-term annual variations in vegetation and snow-hydroclimatic factors were compared to understand the trend pattern. Further, spatiotemporal loss and gain in forest and grassland have been estimated for different time periods (i.e., 2001–2010, 2010–2019, and 2001–2019) in the study area.

### 3.5. Geostatistical Analysis

The non-parametric Mann–Kendall trend test (M-K test) [44] has been used in the present study to calculate a statistically significant linear trend in the forest, grassland, and snow-hydroclimatic factors at various significance levels. This is a rank correlation statistic test that compares the number of discordances observed to the value of the same quantity anticipated from a random series. The World Meteorological Organization has recommended the M-K approach for assessing trends in environmental data time series. This test compares each value in the time series against the others that remain, always in sequential order. The number of times the remaining terms are bigger than the one under consideration is counted. The M-K statistic ($S$) is calculated as follows:

$$S = \sum_{i=2}^{n} \sum_{j=1}^{i-1} sign\left(x_i - y_j\right) \tag{1}$$

where $n$ is the length of data, $x_i$ and $y_j$ are the annual values in years, and the function $sign\left(x_i - y_j\right)$ is calculated by:

$$sign\left(x_i - y_j\right) = \begin{cases} 1, \ if \ \left(x_i - y_j\right) > 0, \\ 0, \ if \ \left(x_i - y_j\right) = 0, \\ -1, \ if \ \left(x_i - y_j\right) < 0, \end{cases} \tag{2}$$

A positive or negative $S$ value denotes an increasing or decreasing trend in the data. If the number of data values is 10 or more, the $S$ statistics approach normal distribution, and the test is completed with the mean $E(S)$ and variation $Var(S)$, as shown in the equations below.

$$E(S) = 0 \tag{3}$$

$$Var(S) = \frac{n(n-1)(2n+5) - \sum_{i=1}^{m} t_i(t_i - 1)(2t_i + 5)}{18} \tag{4}$$

When the sample size is more than 10, a tied group (m) collects data with the same value. $t_i$ is the number of data points in the $i$th tied group. The normal Z test statistic can be estimated using the below equation.

$$Z = \begin{cases} \frac{S-1}{\sqrt{Var(S)}}, \ if \ S > 0 \\ 0, \ if \ S = 0 \\ \frac{S+1}{\sqrt{Var(S)}}, \ if \ S < 0 \end{cases} \tag{5}$$

The Z value is used to estimate the statistical significance of a trend. A positive Z number implies an increasing trend, while a negative value suggests a decreasing trend. In this, the two-tailed test is employed for four distinct significance levels: 0.1, 0.05, 0.01, and 0.001. The significance threshold of 0.001 indicates that there is a 0.1% chance that the values xi are from a random distribution and that with that probability, we make a mistake when rejecting $H_O$ (null hypostasis) of no trend. Thus, a significance level of 0.001 indicates that the occurrence of a monotonic trend is quite likely. The significance threshold of 0.1 indicates that there is a 10% chance that we will make a mistake when rejecting $H_0$. The M-K trend test has been widely used and accepted in hydroclimatic and snow time-series data analysis [45,46].

The sensitivity of forest and grassland for different snow-hydroclimatic factors has been estimated using Pearson's correlation coefficient (r). It calculates the linear relationship between two variables, and its value ranges from −1 to +1 [47]. Positive correlation coefficient values signify a propensity for one variable to increase or decrease along with another, whereas negative correlation coefficient values signify a propensity for one variable



to be linked with a decrease in values with another and vice versa. If there is no linear connection between the variables, the correlation coefficient is 0.

In addition, p-values were computed and the significance threshold of 0.05 was used to accept (reject) the null hypothesis that the correlation coefficient is zero. If the estimated *p*-value is less than 0.05, the null hypothesis $H_O$ (no correlation) is rejected, and vice versa; if the *p*-value is more than 0.05, the correlation is statistically insignificant. Generally, a *p*-value of 0.05 or lower is considered significant [48]. A flow diagram of the overall methodology adopted in this study is shown in Figure 2.

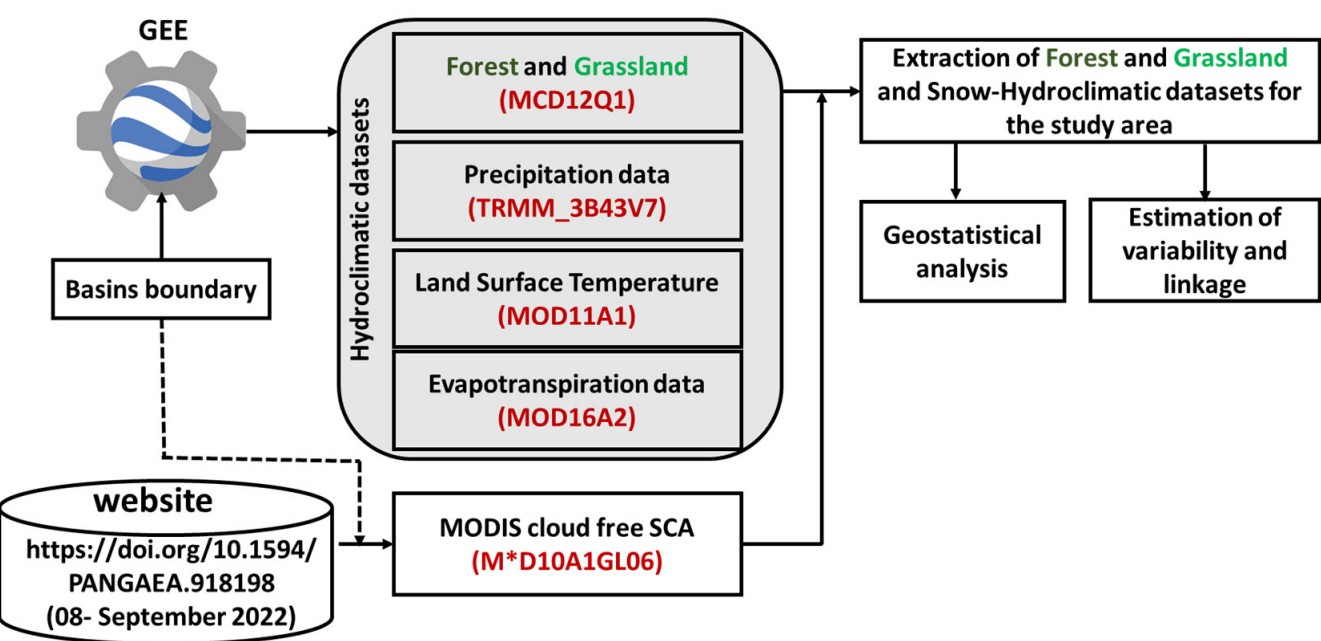

**Figure 2.** Methodology adopted in this study.

## 4. Results & Analysis

### 4.1. Annual Spatiotemporal Variation in Forest and Grassland

Figures 3–5 illustrate the annual spatiotemporal variation of forest and grassland in the study area during the data period from 2001 to 2019. The highest and lowest forest cover percentage was observed as ~19.5% (2001) and 13% (2013), respectively, in the Beas basin. However, the Chandra and Bhaga basins do not have forest cover because in the Western Himalayas, forest persists below 3000 m (a.s.l), and above 3000 m (a.s.l.), grassland patches exist in valley regions. The highest and lowest grassland percentage was observed as ~82.8% (2019) and 74.9% (2001) in the Beas basin, ~21.1% (2018) and 18.4% (2009) in the Chandra basin, and ~8% (2011) and 9.3% (2018) in Bhaga basin. The analysis also exhibits that spatial variability in forest and grassland is directly related to the elevation variation because, at lower elevations, forest and grassland have a highly covered area as compared to higher elevations. The spatiotemporal variation of forest and grassland has been studied over the Beas basin and the spatiotemporal variation of grassland has been studied over the Chandra and Bhaga basins.

### 4.2. Inter- and Intra-Annual Variation of Snow-Hydrolytic Factors

Figure 6a–d shows the inter- and intra-annual variability of PPT, ET, LST, and SCA in the Beas basin for the 19-year period (i.e., from 2001 to 2019). From Figure 6a, the highest and lowest PPT have been observed in the monsoon and post-monsoon seasons, respectively. Overall PPT variability in the Beas basin ranges from 2.2 to 336 mm, whereas, in the winter, pre-monsoon, monsoon, and post-monsoon seasons, it varied from 33.6 to 238.2 mm, 13.9 to 271 mm, 29.9 to 336 mm, and 2.2 to 78.9 mm, respectively, during the data period. The average monthly maximum PPT was observed in the study area to be

170.24 mm in July, 157.09 mm in August, and 130.99 mm in September. The highest and lowest mean annual PPT were found in the years 2018 (108.66 mm) and 2009 (67.02 mm), respectively. From Figure 6b, the highest and lowest ET is observed in the monsoon and winter seasons, respectively. Overall ET variability in the Beas basin ranges from 13.6 to 101.7 mm, whereas, in the winter, pre-monsoon, monsoon, and post-monsoon seasons, it varied from 15.7 to 35.9 mm, 29.3 to 92.1 mm, 48.7 to 101.7 mm, and 13.6 to 44.7 mm, respectively, during the data period. The average monthly maximum ET was found to be 80.77 mm in July, 82.89 mm in August, and 63.44 mm in September in the study area. The highest and lowest mean annual PPT were found in the years 2015 (55.3 mm) and 2002 (37.8 mm), respectively.

From Figure 6c, the highest and lowest LST have been observed in the pre-monsoon and winter seasons, respectively. Overall LST variability in the Beas basin ranges from −6.8 to 18.8 °C, whereas, in the winter, pre-monsoon, monsoon, and post-monsoon seasons, it varies from −6.8 to 4.9 °C, 2.6 to 18.8 °C, 10.1 to 18.0 °C, and −0.9 to 16.4 °C, respectively, during the data period. The average monthly LST in the monsoon season was observed in the study area to be 14.3 °C in July, 13.6 °C in August, and 15.4 °C in September. The maximum and minimum mean annual LST were found in the years 2015 (9.4 °C) and 2019 (7.6 mm), respectively. From Figure 6d, the highest and lowest SCA have been observed in winter and monsoon seasons, respectively. Overall SCA variability in the Beas basin ranges from 7.6 to 86.1%, whereas, in the winter, pre-monsoon, monsoon, and post-monsoon seasons, it varied from 59.2 to 86.1%, 18.3 to 84.5%, 10.3 to 38.9%, and 7.6 to 73.6%, respectively, during the data period. The average monthly SCA in the monsoon season was observed to be 23.9% in July, 17.5% in August, and 17.3% in September in the study area. The maximum and minimum annual SCA were found in the years 2014 (50.6%) and 2016 (35.8%), respectively.

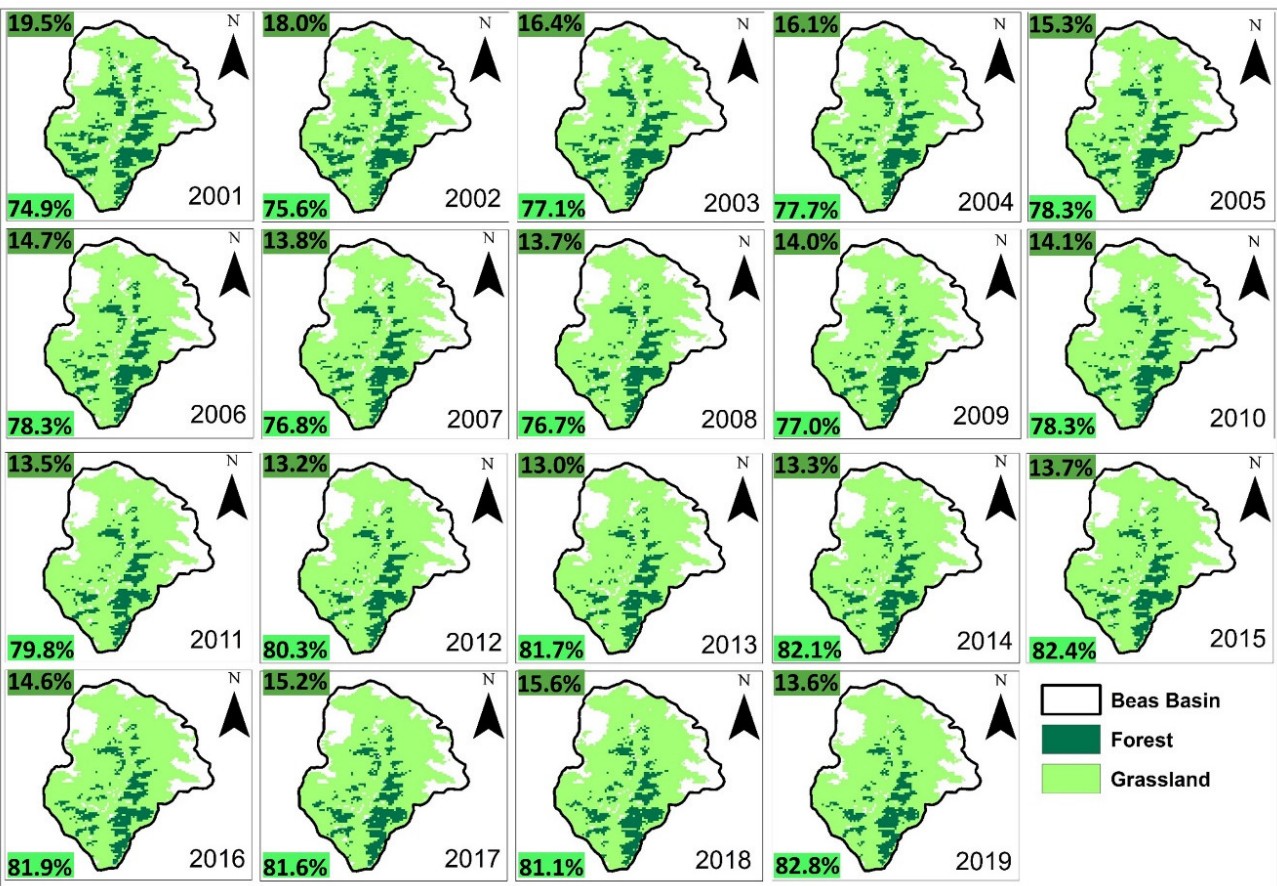

**Figure 3.** Annual spatiotemporal variation of forest and grassland in the Beas basin from 2001 to 2019.



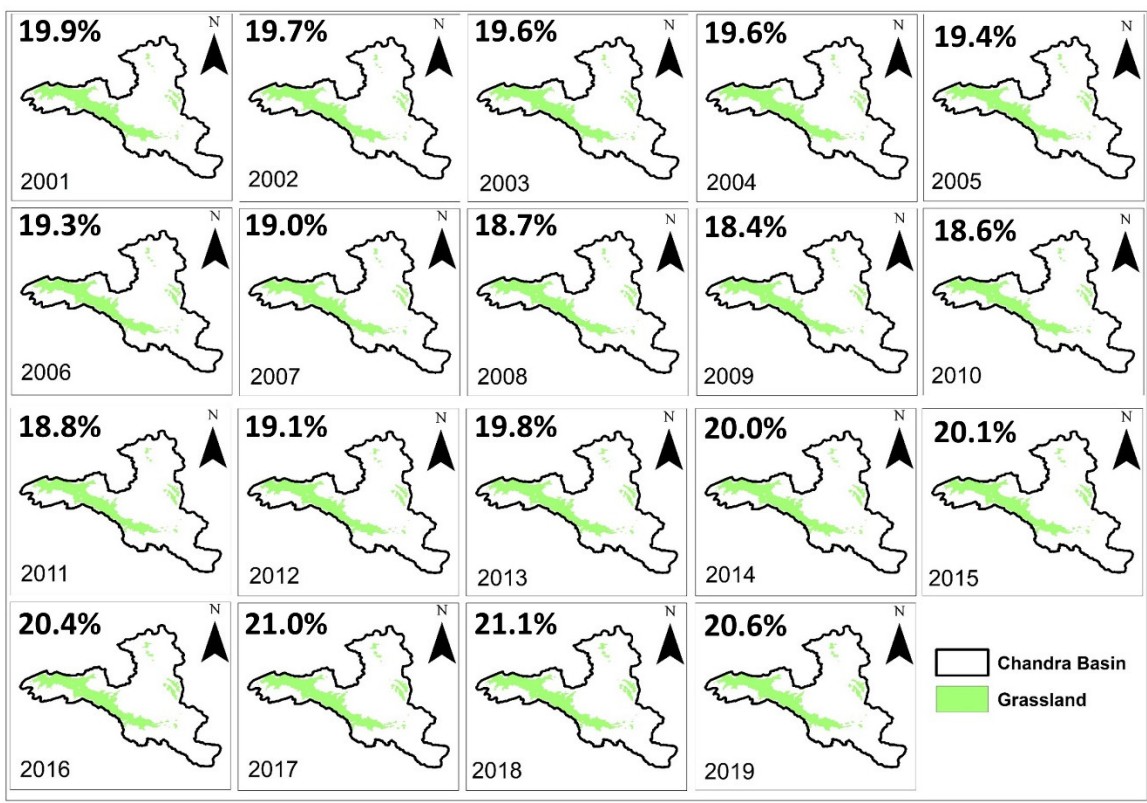

**Figure 4.** Annual spatiotemporal variation of grassland in the Chandra basin from 2001 to 2019.

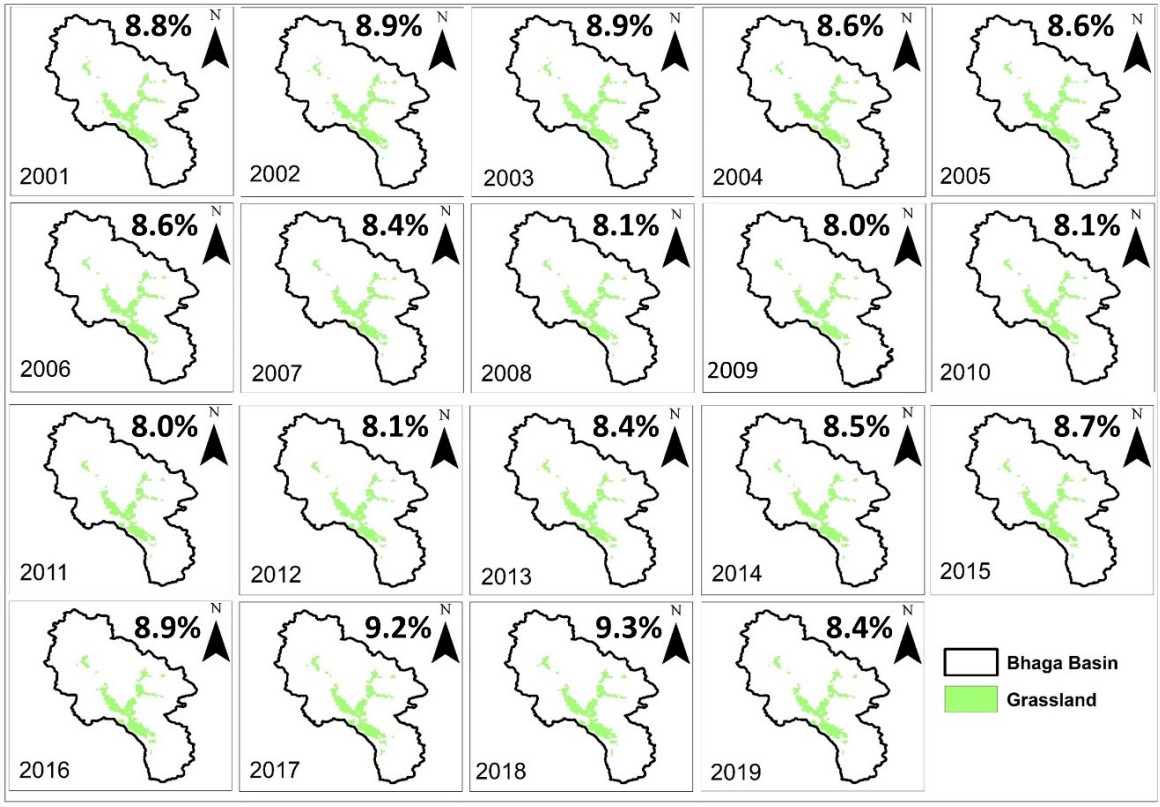

**Figure 5.** Annual spatiotemporal variation of grassland in the Bhaga basin from 2001 to 2019.

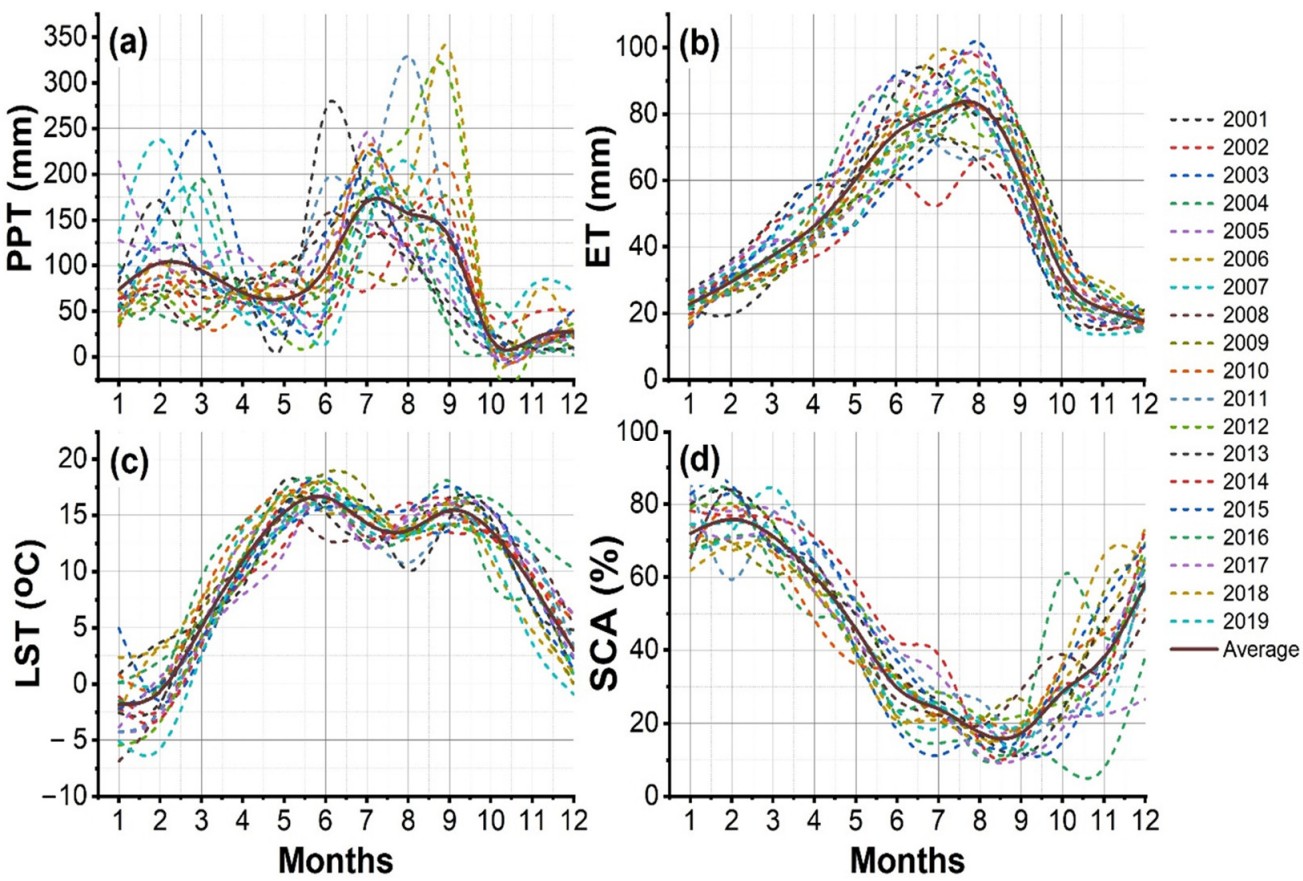

**Figure 6.** Inter- and intra-annual variation of (**a**) PPT, (**b**) ET, (**c**) LST, and (**d**) SCA in the Beas basin from 2001 to 2019.

Figure 7a–d shows the inter- and intra-annual variability of PPT, ET, LST, and SCA in the Chandra basin for the 19-year period (i.e., from 2001 to 2019). From Figure 7a, the highest and lowest PPT have been observed in the monsoon and post-monsoon seasons, respectively. Overall PPT variability in the Chandra basin ranges from 1.5 to 311.9 mm, whereas, in the winter, pre-monsoon, monsoon, and post-monsoon seasons, it varied from 34.3 to 252.7 mm, 24.1 to 206.7 mm, 35.1 to 311.9 mm, and 1.5 to 93.5 mm, respectively, during the data period. The average monthly maximum PPT was reported 114.7 mm in July, 101.2 mm in August, and 110.6 mm in September in the study area. The highest and lowest mean annual PPT were found in the year 2019 (100.9 mm) and 2004 (58.9 mm), respectively. From Figure 7b, the highest and lowest ET have been observed in the monsoon and winter seasons, respectively. Overall, ET variability in the Chandra basin ranges from 20.2 to 100.6 mm, whereas, in the winter, pre-monsoon, monsoon, and post-monsoon seasons, it varied from 20.2 to 46.1 mm, 41.1 to 101.1 mm, 37.9 to 100.63 mm, and 47.8 to 47.8 mm, respectively, during the data period. The average monthly maximum ET was observed in the study area to be 85.8 mm in July, 81.1 mm in August, and 55.4 mm in September. The highest and lowest mean annual PPT were found in the years 2013 (60.5 mm) and 2001 (48.8 mm), respectively. From Figure 7c, the highest and lowest LST has been reported in the pre-monsoon and winter seasons, respectively. Overall LST variability in the Chandra basin ranges from −19.7 to 16.5 °C, whereas, in the winter, pre-monsoon, monsoon, and post-monsoon seasons, it varies from −19.7 to −9.2 °C, −9.8 to 12.9 °C, 5.1 to 16.5 °C, and −16.6 to 1.03 °C, respectively, during the data period. The average monthly LST in the monsoon season was 13.5 °C in July, 13.8 °C in August, and 11.3 °C in September in the study area. The maximum and minimum mean annual LST were found in the years 2016 (3.0 °C) and 2019 (−1.9 °C), respectively. From Figure 7d, the highest and lowest

SCA is found in winter and monsoon seasons, respectively. Overall SCA variability in the Beas basin ranges from 42 to 100%, whereas, in the winter, pre-monsoon, monsoon, and post-monsoon seasons, it varied from 66 to 100%, 67 to 100%, 42 to 89%, and 52 to 99%, respectively, during the data period. The average monthly SCA in the monsoon season was observed to be 59.4% in July, 51% in August, and 66% in September in the study area. The maximum and minimum annual SCA were found in the years 2009 (89.9%) and 2016 (74.4%), respectively.

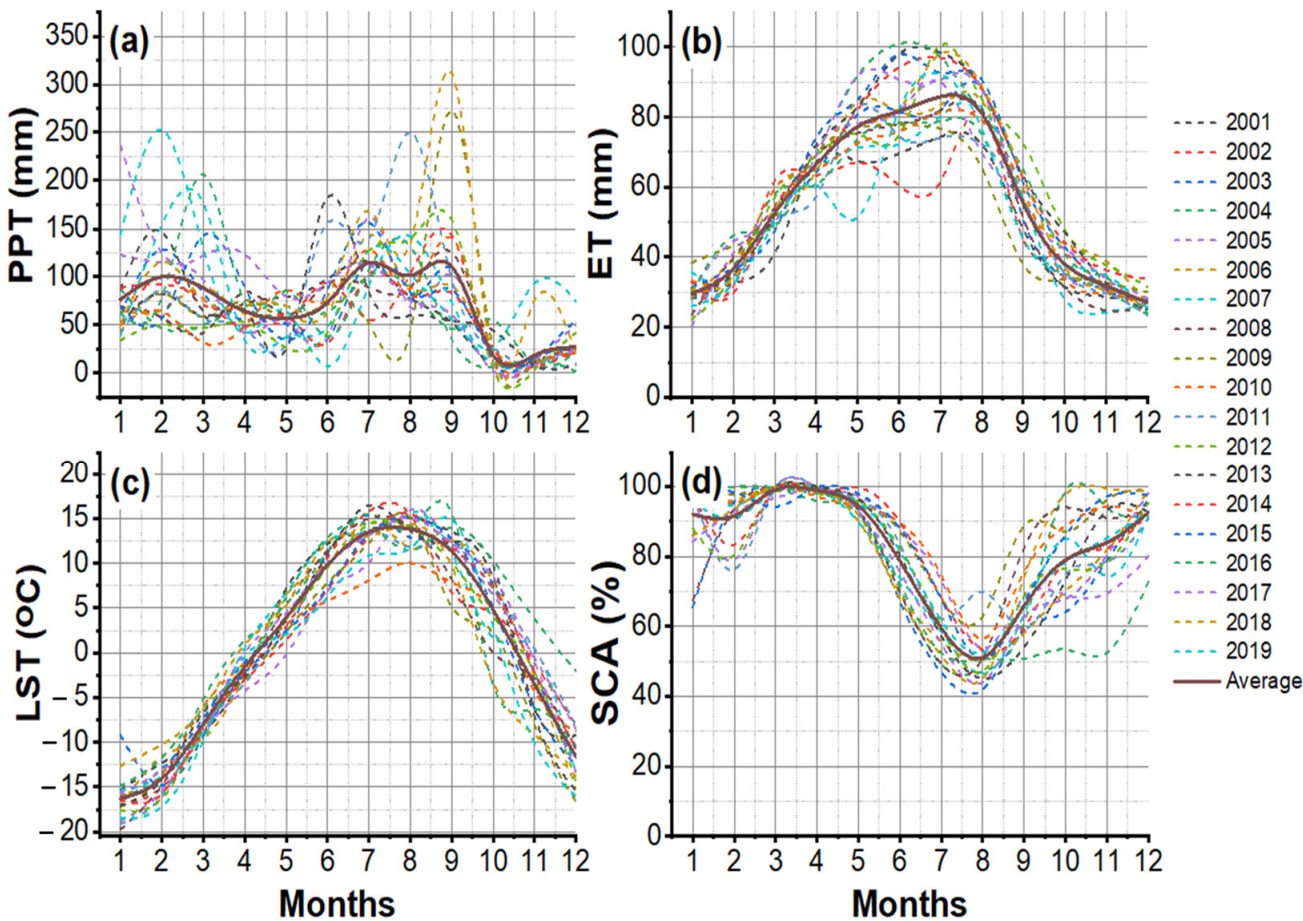

**Figure 7.** Intra- and inter-annual variation of (**a**) PPT, (**b**) ET, (**c**) LST, and (**d**) SCA in the Chandra basin from 2001 to 2019.

Figure 8a–d shows the inter- and intra-annual variability of PPT, ET, LST, and SCA in the Bhaga basin for the 19-year period (i.e., from 2001 to 2019). From Figure 8a, the highest and lowest PPT has been reported in the monsoon and post-monsoon seasons, respectively. Overall PPT variability in the Bhaga basin ranges from 0.9 to 317.1 mm, whereas, in the winter, pre-monsoon, monsoon, and post-monsoon seasons, it varied from 32.1 to 249.8 mm, 9.5 to 193.6 mm, 29.2 to 317 mm, and 0.7 to 108.1 mm, respectively, during the data period. The average monthly maximum PPT was 90.8 mm in July, 87.4 mm in August, and 100.8 mm in September in the study area. The highest and lowest mean annual PPT were found in the year 2018 (88.9 mm) and 2004 (53.4 mm), respectively. From Figure 8b, the highest and lowest ET have been in the monsoon and winter seasons, respectively. Overall ET variability in the Bhaga basin ranges from 17.7 to 103.9 mm, whereas, in the winter, pre-monsoon, monsoon, and post-monsoon seasons, it varied from 17.7 to 42.8 mm, 42.1 to 103.9 mm, 36.4 to 96.0 mm, and 22.0 to 46.6 mm, respectively, during the data period. The average monthly maximum ET was 80.3 mm in July, 75.3 mm in August, and 52 mm in September in the study area. The highest and lowest mean annual PPT were found in the

years 2016 (59.3 mm) and 2001 (45.4 mm), respectively. From Figure 8c, the highest and lowest LST have been observed in the monsoon and winter seasons, respectively. Overall LST variability in the Bhaga basin ranges from $-20.2$ to $17.8\ °C$, whereas, in the winter, pre-monsoon, monsoon, and post-monsoon seasons, it varies from $-20.2$ to $-10.4\ °C$, $-9.7$ to $14.6\ °C$, $7.2$ to $17.8\ °C$, and $17$ to $10.7\ °C$, respectively, during the data period. The average monthly LST in the monsoon season was $14.1\ °C$ in July, $14.4\ °C$ in August, and $12.1\ °C$ in September in the study area. The maximum and minimum mean annual LST were found in the years 2016 ($3.4\ °C$) and 2019 ($-1.6\ °C$), respectively. From Figure 8d, the highest and lowest SCA have been observed in winter and monsoon seasons, respectively. Overall SCA variability in the Bhaga basin ranges from 38.2 to 100%, whereas, in the winter, pre-monsoon, monsoon, and post-monsoon seasons, it varied from 66.3 to 100%, 70.5 to 100%, 38.2 to 85%, and 53.4 to 99.4%, respectively, during the data period. The average monthly SCA in the monsoon season was 60.1% in July, 49.9% in August, and 65.8% in September in the study area. The maximum and minimum annual SCA were found in the years 2009 (90.6%) and 2016 (73.6%), respectively.

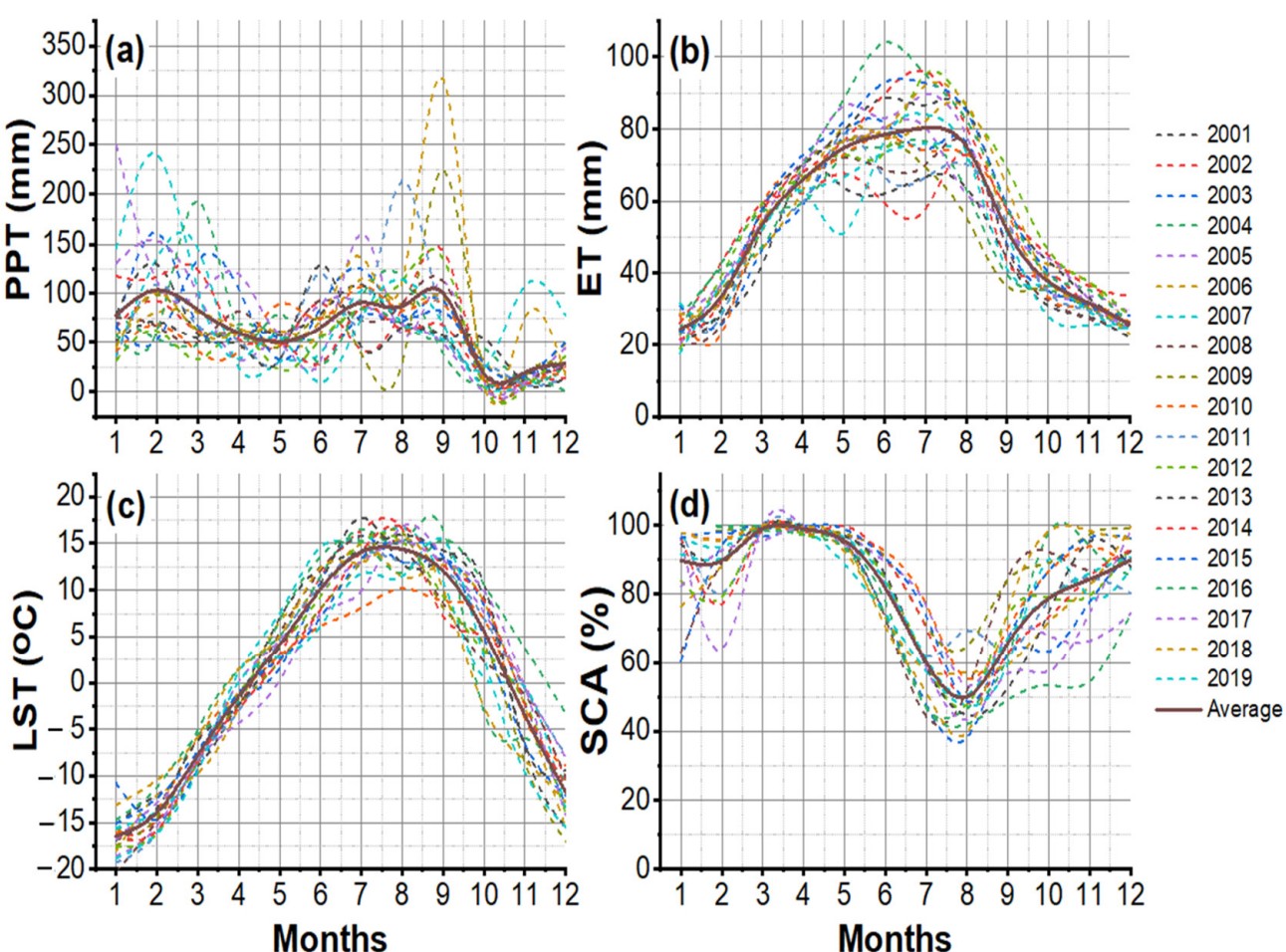

**Figure 8.** Intra- and inter-annual variation of (**a**) PPT, (**b**) ET, (**c**) LST, and (**d**) SCA in the Bhaga basin during 2001–2019.

### 4.3. Temporal Variation and Trends Analysis

The mean annual temporal variation of vegetation and snow-hydroclimatic factors and statistical significance ($\alpha$) of the trend in the Beas basin during the data period (2001–2019) are shown in Figure 9. A significant decreasing trend of ~214 ha/yr. has been observed in the forest at $\alpha = 0.01$ from 2001 to 2019. However, a significantly increasing trend has been observed in the grassland (459 ha/yr.), ET (0.76 mm/yr.), and PPT (1.4 mm/yr.) at $\alpha = 0.001$, $\alpha = 0.001$, and $\alpha = 0.01$, respectively. A non-significant decreasing and increasing

trend has been observed in LST (0.03 K/yr.) and snow cover area (120 ha/yr.) in the study area during the data period.

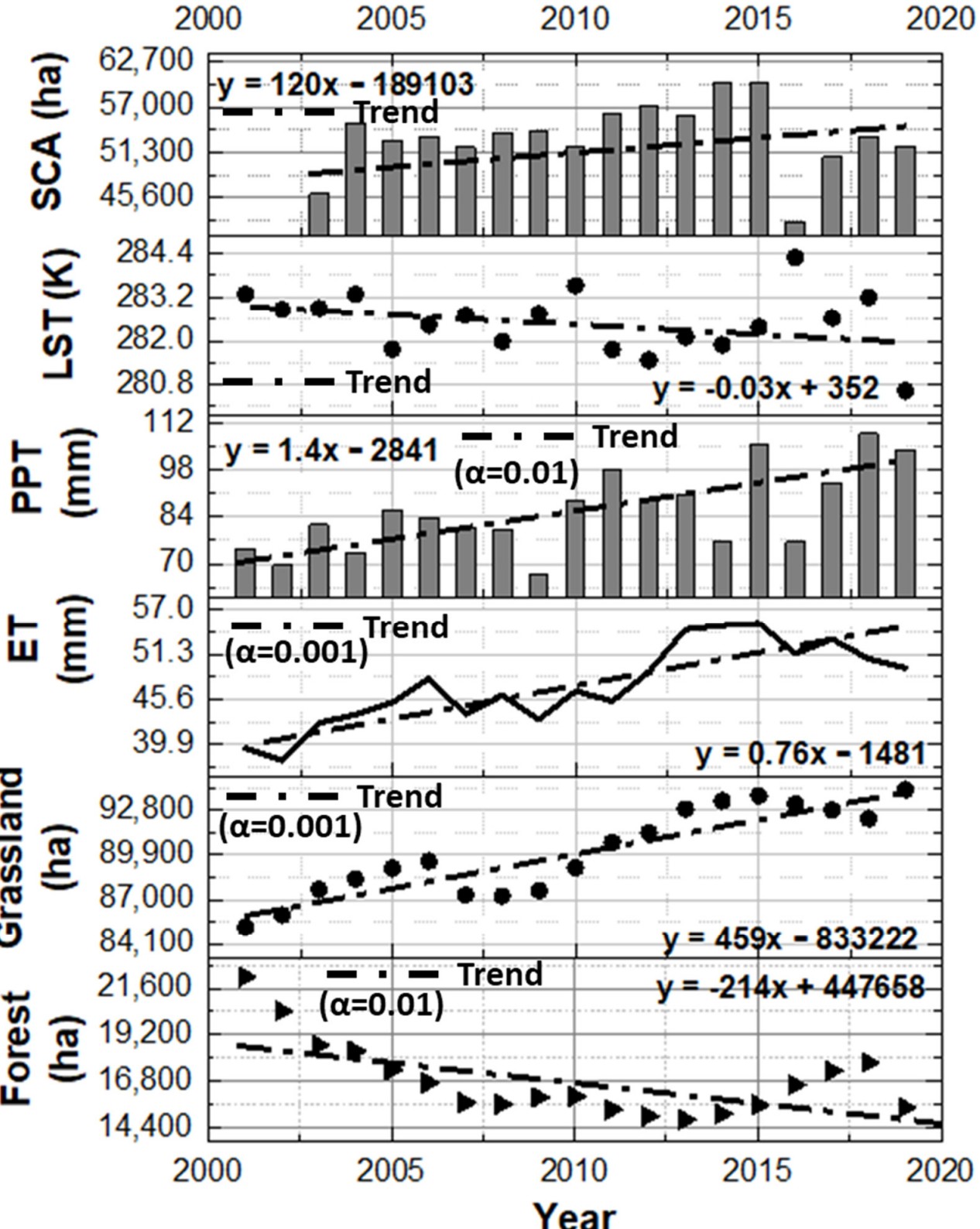

**Figure 9.** Temporal variation of annual mean forest cover, grassland, ET, PPT, LST, and SCA and statistical significance (α) of the trend in the Beas basin during 2001–2019.

The mean annual temporal variation of grassland and snow-hydroclimatic factors and statistical significance ($\alpha$) of the trend in the Chandra basin during the data period (2001–2019) are shown in Figure 10. A significantly increasing trend has been observed in the grassland (176.9 ha/yr.), ET (0.39 mm/yr.), and PPT (1.6 mm/yr.) at $\alpha = 0.1$, $\alpha = 0.05$, and $\alpha = 0.01$, respectively. However, a non-significant decreasing and increasing trend has been observed in LST (0.02 K/yr.) and SCA (236 ha/yr.) in the study area during the data period.

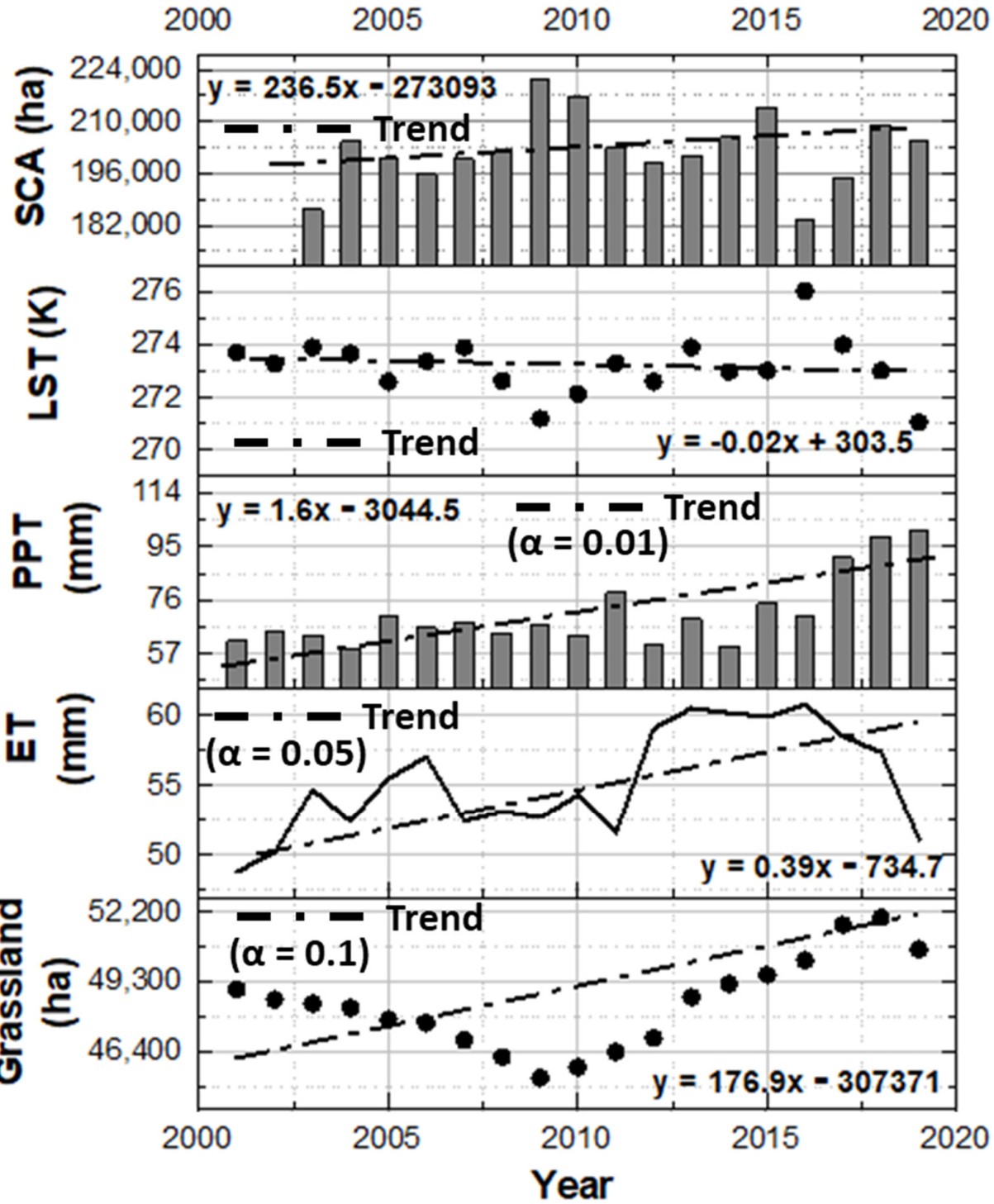

**Figure 10.** Temporal variation of annual mean grassland, ET, PPT, LST, and SCA and statistical significance ($\alpha$) of the trend in the Chandra basin during 2001–2019.

The mean annual temporal variation of grassland and snow-hydroclimatic factors and statistical significance ($\alpha$) of the trend in the Bhaga basin during the data period (2001–2019) are shown in Figure 11. A significantly increasing trend has been observed in the ET (0.44 mm/yr.) and PPT (1.6 mm/yr.) at $\alpha$ = 0.05. However, a non-significant decreasing trend has been observed in LST (0.01 K/yr.) and an increasing trend in grassland (9.1 ha/yr.) and SCA (120.2 ha/yr.) in the study area during the data period. During the data period, the overall M-K trend and its statistical significance for the forest, grassland, and snow-hydroclimatic factor in HP basins is given in Table 2.

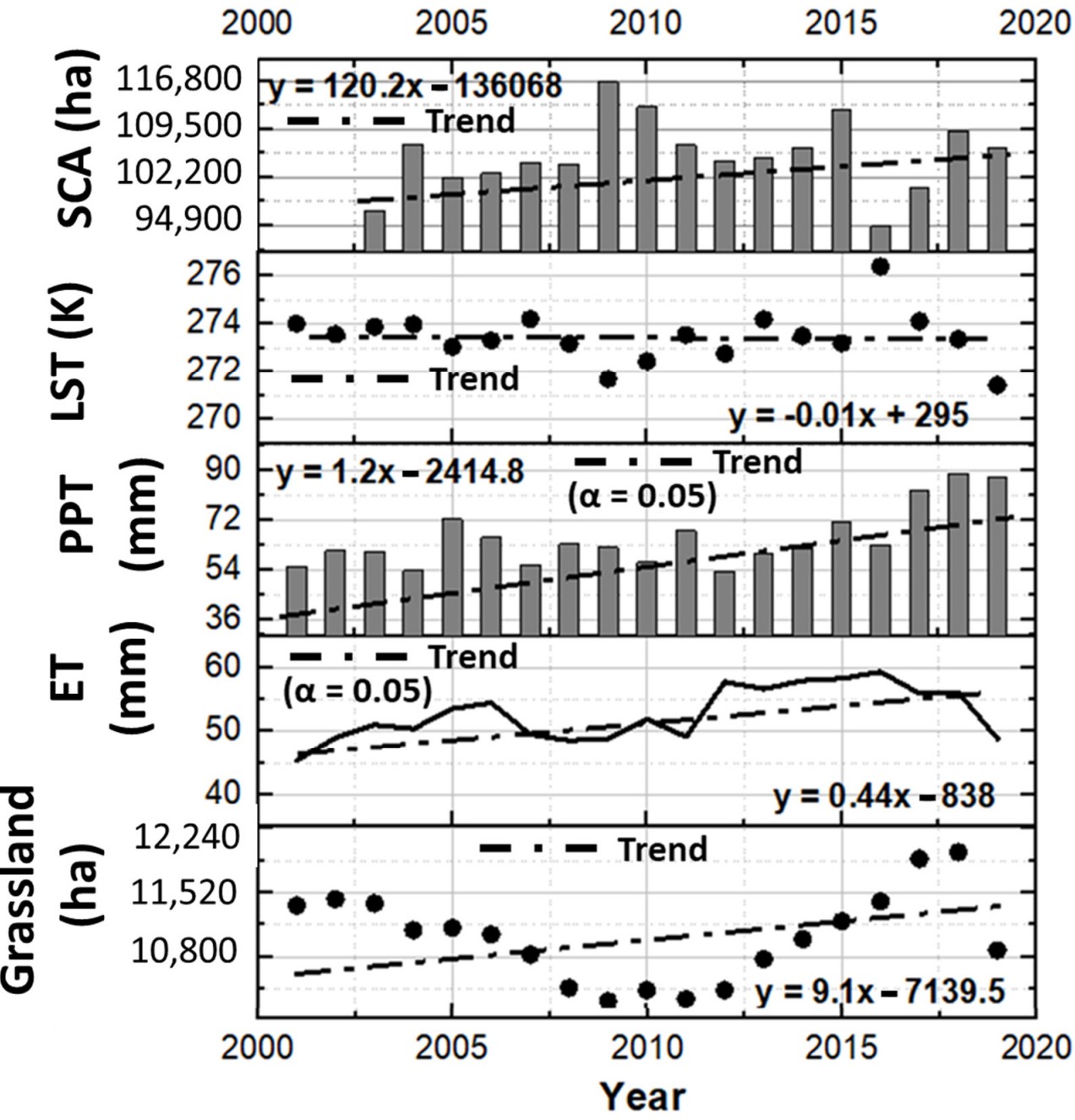

**Figure 11.** Temporal variation of annual mean grassland, ET, PPT, LST, and SCA, and statistical significance ($\alpha$) of the trend in the Bhaga basin during 2001–2019.

**Table 2.** M-K trend and its statistical significance for the forest, grassland, and snow-hydroclimatic factor in HP basins during the data period (Note: here, ***, **, *, and + represent significance levels at 0.001, 0.01, 0.05, and 0.1, respectively).

| Trend | Beas Basin | | | Chandra Basin | | | Bhaga Basin | | |
|---|---|---|---|---|---|---|---|---|---|
| | Slope | Z-Value | Significant | Slope | Z-Value | Significant | Slope | Z-Value | Significant |
| **Forest (ha/yr)** | −214 | −2.59 | ** | - | - | - | - | - | - |
| **Grassland (ha/yr)** | 459 | 4.20 | *** | 176.9 | 1.75 | + | 9.1 | −0.04 | |
| **ET (mm/yr)** | 0.76 | 3.85 | *** | 0.39 | 2.31 | * | 0.44 | 2.38 | * |
| **PPT (mm/yr)** | 1.4 | 2.87 | ** | 1.6 | 2.66 | ** | 1.2 | 2.24 | * |
| **LST (K/yr)** | −0.03 | −1.12 | | −0.02 | −0.28 | | −0.01 | −0.63 | |
| **SCA (ha/yr)** | 120 | 0.62 | | 236.5 | 0.70 | | 120.2 | 0.87 | |

### 4.4. Spatiotemporal Change in Forest and Grassland

Figure 12 shows the spatiotemporal change in forest and grassland in the study area from 2001 to 2019. From Figure 12a–c, during 2001–2019 in the Beas basin, spatiotemporal forest loss and gain were observed at ~6.6% and 1.6%, respectively. However, grassland loss and gain were ~2.9% and 9.4%, respectively. From 2001 to 2010, the spatiotemporal forest loss and gain were 5.6% and 1.1%, respectively. However, 3.8% and 6.6% loss and gain were in grassland, respectively. During 2010–2019 spatiotemporal forest loss and gain were observed at ~1.9% and 1.6%, respectively. However, in grassland, loss and gain were observed at ~2.1% and 5.8%, respectively. Compared to 2001–2010, the analysis shows a high spatiotemporal gain and a low loss from 2010 to 2019 in the Beas basin.

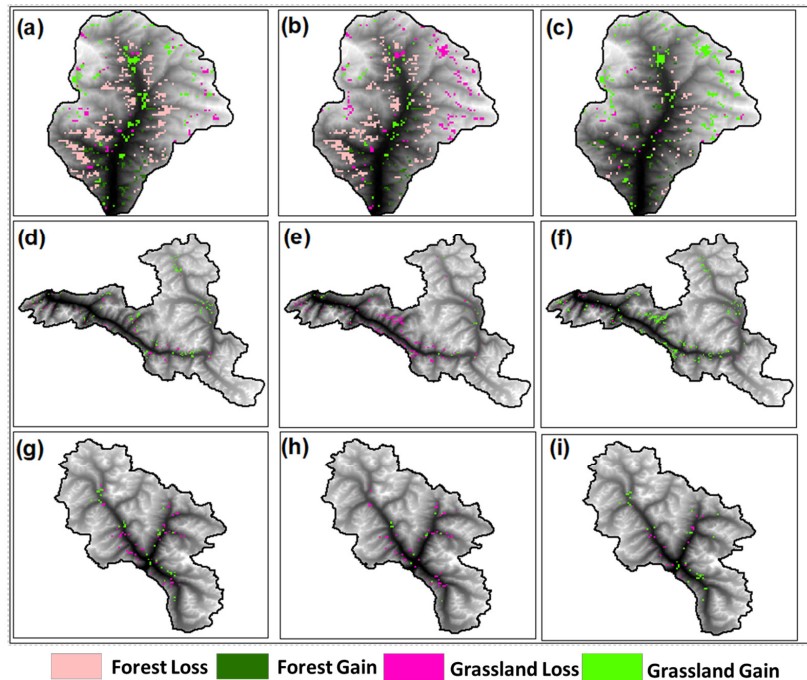

**Figure 12.** Spatiotemporal gain and loss of forest and grassland in the Beas basin (**a**) during 2001–2019, (**b**) during 2001–2010, and (**c**) during 2010–2019. Grassland gain and loss in the Chandra basin (**d**) during 2001–2019, (**e**) during 2001–2010, and (**f**) during 2010–2019. Grassland gain and loss in the Bhaga basin (**g**) during 2001–2019, (**h**) during 2001–2010, and (**i**) during 2010–2019.

From Figure 12d-e, during 2001–2019 in the Chandra basin, spatiotemporal grassland loss and gain were observed at ~0.59% and 1.16%, respectively. During 2001–2010, spatiotemporal grassland loss and gain were observed at ~1.38% and 0.28%, respectively. During 2010–2019, the spatiotemporal grassland loss and gain were observed at ~0.17% and 1.84%, respectively. In comparison to 2001–2010, the analysis shows a high spatiotemporal gain and a low loss in grassland from 2010 to 2019.

From Figure 12f–i, during 2001–2019 in the Bhaga basin, spatiotemporal grassland loss and gain were observed at ~0.92% and 0.60%, respectively. During 2001–2010, the spatiotemporal grassland loss and gain were observed at ~0.84% and 0.23%, respectively. During 2010–2019, the spatiotemporal grassland loss and gain were observed at ~0.31% and 0.60%, respectively. In the Bhaga basin, very little gain and loss have been observed as compared to the Chandra and Beas basins.

### 4.5. The Relationship between Forest and Grassland with Snow-Hydroclimatic Factors

The annual Pearson correlation coefficient values, *p*-value, and statistical significance at $\alpha = 0.05$ level between forest and grassland with snow-hydroclimatic factors in the study area during the data period are shown in Figure 13. In the Beas basin, a strong negative correlation (r = −0.65) between forest and evapotranspiration and a weak negative correlation (r = −0.37) between forest and precipitation have been observed during the data period. A significant correlation ($p < 0.05$) is observed between the forest, ET, LST, and SCA. However, an insignificant correlation ($p > 0.05$) is observed between the forest and LST. Furthermore, a strong positive correlation (r = 0.99) between grassland and ET and a weak positive correlation (r = 0.18) between grassland and SCA have been observed during the data period. A significant correlation ($p < 0.05$) is observed between the grassland, ET, and PPT. However, an insignificant correlation ($p > 0.05$) is observed between the grassland LST and SCA.

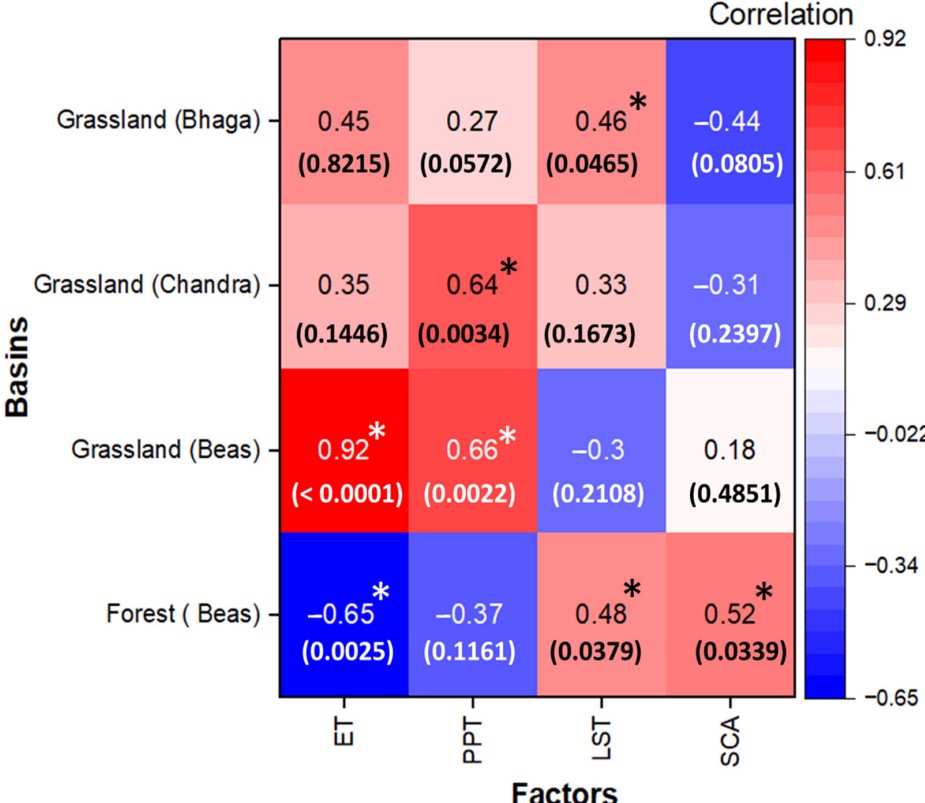

**Figure 13.** Heat map of annual Pearson correlation coefficient, *p*-value, and * represents statistical significance at the 0.05 level for forest and grassland with ET, PPT, LST, and SCA in the study area.

In the Chandra basin, a strong positive correlation (r = 0.64) between grassland and PPT and a weak negative correlation (r = −0.31) between grassland and SCA has been observed during the data. A significant correlation ($p < 0.05$) is observed between the grassland and PPT. However, an insignificant correlation ($p > 0.05$) is observed between the grassland, ET, LST, and SCA.

In the Bhaga basin, a strong positive correlation (r = 0.45) between grassland and ET and a weak positive correlation (r = 0.27) between grassland and PPT has been observed during the data period. A significant correlation ($p < 0.05$) is observed between the grassland and LST. However, an insignificant correlation ($p > 0.05$) is observed between the grassland, ET, PPT, and SCA.

## 5. Discussion

Long-term forest and grassland trend analysis is critical for monitoring vegetation conditions and identifying the hotspot locations for future research. Such data could be used to identify areas prone to vegetation loss in order to prevent further land degradation and hazards. Not only do negative trends necessitate further investigation, but the areas with positive trends, particularly those with large slope magnitudes, should also be considered. This is due to the possibility of invasive vegetation growth or arbitrary agricultural expansion, both of which could have negative environmental consequences. As a result, the trend analysis findings for the period 2001–2019 could aid scholars and decision-makers in conducting research to identify driving factors and enacting effective legislation.

In the present study, the spatiotemporal variability of vegetation (i.e., greenness and forest) and associated snow-hydroclimatic factors (i.e., SCA, LST, PPT, and ET) within the three Basins of HP (i.e., Beas, Chandra, and Bhaga) has been analyzed. The study area has a temperate climate in the lower Himalayan range. The basin is warmer in summers and cooler in winters under these climatic conditions. The highest amount of PPT was observed in July, August, and September, as seen in Figures 6a, 7a, and 8a. Additionally, this analysis is consistent with previous research conducted in the Indian subcontinent's Himalayan region [26,28]. Furthermore, throughout the data period (2091–2019), the ET had a lower value than the PPT in most of the months in the study area. This demonstrates that the study area basins are dominated by energy-limited environments [49,50]. Temperature showed the least variation in the last 19 years compared to PPT (Figures 6a, 7a, and 8a), ET (Figures 6b, 7b, and 8b), and SCA (Figures 6d, 7d, and 8d); however, there were seasonal fluctuations (Figures 6c, 7c, and 8c). This could be related to changes in PPT and ET within this region throughout time. The vegetation pattern changed across the lower Himalayan range, altering the feedback mechanisms between climate–vegetation interactions [51,52].

The vegetation over the Himalayan basins was mainly explored using vegetation indices, such as NDVI, EVI, etc., [26]. However, for the first time, this research has used vegetation types, namely forest and grassland, using yearly MODIS land cover product data with superior accuracy. During the data period, forest and grassland revealed a decreasing and increasing trend, respectively, in the study area. The annual mean forest decreased from 22,000 ha (2001) to 15,425 ha (2019) in the Beas basin. However, annual mean grassland increased from 85,200 ha (2001) to 94,100 ha (2019) in the Beas basin, 48,975 ha (2001) to 50,625 ha (2019) in the Chandra basin, and 11,375 ha (2001) to 11,975 ha (2018) in Bhaga basin. This demonstrates that in the last 19 years, grassland has increased in the study area; however, forest has decreased in the Beas basin. This trend is corroborated by prior research in similar environments [26,28,53].

In the study area, the forest seems to be well-correlated with ET, while grassland appears to be well-correlated with ET and PPT. Kumari et al. [26] have reported that EVI has a higher correlation with ET and precipitation than NDVI in the Uttarakhand Basin. Analysis reveals that the magnitudes of vegetation phenology (i.e., forest and grassland) correlated strongly with the magnitudes of ET and PPT. ET and PPT are more sensitive in temperate climatic locations, as is clear from the significant association between the forest and grassland changes in vegetation phenology. These findings are consistent with prior

research that found a substantial correlation between the seasonality of satellite-derived vegetation indices and hydroclimatic (e.g., ET) parameters [54].

## 6. Conclusions

The Indian Western Himalaya ecosystem (between the tree line and the snow line) is an underappreciated area. Yet, knowledge of vegetation distribution, rates of change, and vegetation interactions with snow-hydroclimatic elements is lacking. The main objective of the present study was to understand how vegetation responds to various snow-hydroclimatic factors in the Himalayan ecosystem. Monitoring vegetation in the Himalayan areas is challenging due to the rocky and complicated terrain and very severe climate conditions. As a result, a remote sensing approach is beneficial in studying and monitoring vegetation and evaluating its snow-hydrological response in unmeasured basins. In this study, the three HP Basins (i.e., Beas, Chandra, and Bhaga) that lie in the lower Himalayan ranges are used to understand the long-term spatiotemporal variability of the forest and grassland and its relation with snow-hydroclimatic factors.

Forest and grassland have been estimated from the MODIS product using GEE for the period 2001 to 2019. Large spatiotemporal variability was detected in the Beas basin forest, with the lowest and highest values reported in 2001 (19.5%) and 2013 (13%), respectively, whereas the highest lowest grassland percentages were 82.8% (2019) and 74.9% (2001), respectively, in the Beas basin, 21.1% (2018) and 18.4% (2009) in the Chandra basin, and 8% (2011) and 9.3% (2018) in the Bhaga basin. In addition, current PPT, LST, ET, and SCA satellite data were used to better understand the relation and sensitivity of vegetation with snow-hydroclimatic conditions in the study area. The analysis of forest and grassland shows a strong seasonal control, with seasonal peaks in September and August, respectively.

The correlation between vegetation and snow-hydroclimatic factors shows a high negative correlation between forest and ET in the Beas basin. However, there was a high positive connection between grassland and ET and a modest positive correlation between grassland and SCA. Further, a high positive correlation between grassland and PPT and a slight negative correlation between grassland and SCA was observed in the Chandra basin, followed by a high positive connection between grassland and ET and a modest positive correlation between grassland and PPT in the Bhaga basin. This shows that variability in vegetation is highly dependent on basin climatology (i.e., ET and PPT) and altitude. The main outcome of this study in terms of Spatiotemporal loss and gain in forest and grassland shows that in the Bhaga basin, very little gain and loss have been observed as compared to the Chandra and Beas basins. The findings of this study may aid in the protection of the natural environment and the management of water resources in the HP Basin and other high-mountain regions of the Himalayas.

In addition, future research can look at the varied types of vegetation and the periodic large-scale die-offs of certain plant species in the Himalayas. Also, in future research, more accurate and trustworthy high-resolution vegetation maps and seasonal variability may be generated by taking into account high-resolution remote sensing data. As a result, future research could increase vegetation class accuracy by developing a better algorithm for mapping distinct areas. Seasonal vegetation class inventory maps for additional Himalayan regions could also be created. Also, the slope of the land has a significant impact on soil fertility and the amount of water in the soil. On steep slopes, the soil is more prone to erosion, which can lead to a loss of fertile topsoil. This can make it more difficult for plants to grow and can also lead to a decline in soil productivity over time. On the other hand, gentle slopes can be beneficial for soil fertility as they can allow for better water infiltration, which can help to improve soil structure and fertility. This type of analysis has not been considered in this study. Furthermore, future vegetation classification studies could make use of high-resolution satellite photos such as GeoEye, Quick Bird, and WorldView to improve quantitative accuracy. The limitation of the current study is that we have considered only annual LULC products with coarse resolution and limited hydroclimatic factors, which may be overcome in the future using other satellite data.



**Author Contributions:** Conceptualization and Supervision: D.K.S., D.K.G., G.P.P., S.S. and K.K.S.; methodology and writing—original draft preparation: D.K.S., P.J., D.K.G. and S.S.; discussion: D.K.S., D.K.G., P.S.B., S.S. and G.P.P.; review, editing, and visualization: S.S., D.K.S., G.P.P., D.K.G. and V.S.; formal analysis: D.K.S., P.S.B. and P.J. All authors have read and agreed to the published version of the manuscript.

**Funding:** This research received no external funding.

**Data Availability Statement:** Not applicable.

**Conflicts of Interest:** The authors declare no conflict of interest.

## Abbreviations

| | |
|---|---|
| a.s.l. | Above Sea Level |
| ASTER | Advanced Spaceborne Thermal Emission and Reflection Radiometer |
| AWiFS | Advanced Wide Field Sensor |
| CARI | Chlorophyll Absorption Ratio Index |
| DVI | Difference Vegetation Index |
| ET | Evapotranspiration |
| FSI | Forest Survey of India |
| GEE | Google Earth Engine |
| ha | Hectares |
| HP | Himachal Pradesh |
| IGBP | International Geosphere-Biosphere Program |
| IHR | Indian Himalayan Region |
| LST | Land Surface Temperature |
| MCARI | Modified Chlorophyll Absorption Ratio Index |
| M-K | Mann–Kendall |
| MODIS | Moderate Resolution Imaging Spectroradiometer |
| NDVI | Normalized Difference Vegetation Index |
| PPT | Precipitation |
| RSVI | Renormalized Difference Vegetation Index |
| SCA | Snow Cover Area |
| TGDVI | Three-band Gradient Difference Vegetation Index |
| TRMM | Tropical Rainfall Measuring Mission |
| TVI | Triangular Vegetation Index |

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
