# Peer review of "Spatiotemporal Vegetation Variability and Linkage with Snow-Hydroclimatic Factors in Western Himalaya Using Remote Sensing and Google Earth Engine (GEE)"

_remotesensing, doi:10.3390/rs15215239_

Round 1

Reviewer 1 Report

Comments and Suggestions for Authors

Thank you for submitting your work to remote sensing.

Please check the following comments:

Line 58: please provide specific link to access the report.

Second paragraph of the introduction: where is the link with the Himalayan region?

Where is this in the references? Forest Resources Assessment, 2000

Where is this in the references? FSI 2001

How do you overpass the lack of data as stated in line 112?

Rephrase of remove this sentence is too vague "some of the data products available on Google Earth Engine (GEE) have not been thoroughly examined in the literature until present time."

seems line 187 is out of place

rename figures 2a 2b and 2c as figure 2 figure 3 figure 4. Do the same with other figures, usually you can do this when all the figures are in the same frame. As you properly did in Figure 3.

Add a figure including the whole workflow.

Must improve visualization of Figure 6 7 and 8

Fix data period in line 523 

check line 543

Please write conclusion section again, as now, is mainly a summary of the results, in addition, only lines 593-595 are related to the main objective of your work and related to the its title.

There must be more angles to approach in future research, please be more critical

Add in the conclusions the limitations of your actual work

it was not understand during the whole text if there was now a better understanding from using GEE, also if, as you stated at the beginning of your work if GEE has not been used/examined then you must compare the procedure from GEE versus some other workflow. Discuss. 

Author Response

I am revised the manuscript with our best efforts. Really your suggestion were very constructive, which improves lot in my revised manuscript. After this, if anything left, please let us know, we will surely included in the next revision of this manuscript. 

Regards

Dileep (corresponding author)

Reviewer 2 Report

Comments and Suggestions for Authors

The manuscript entitled Spatiotemporal Vegetation Variability and Linkage with Snow Hydroclimatic Factors Using Remote Sensing in Western Himalaya after Major Revision.

In my opinion, the topic of this work is relevant to the journal Remote Sensing MDPI

The topic of the paper is very interesting and important, especially in the analysis of long-term observation and connection between snow and hydroclimatic factors.

Initially, the paper contains the following sections and subsections (i.e., abstract, material, study area, Geospatial Data and Methods , Forest and Grassland , Hydroclimatic Datasets , Method of evaluation , Trend and sensitivity analysis , Results & analysis , Annual spatiotemporal variation in forest and grassland , Inter and inter annual variation of snow-hydrolytic factors , Temporal variation and trends analysis , Spatiotemporal change in forest and grassland , The relationship between forest and grassland with snow-hydroclimatic factors , Discussion conclusions, etc.).

Abstract

The summary section of Abstract it is necessary to extend with minimum one sentence explaining the main findings of this research or how this research can expand knowledge on this important topic. In this part of the paper, it is important to explain the seasonal values of the measured parameters. Also, the methodology used to classify the forest areas should be better explained. At the end of this section, I saw that the authors used two metric units, one is ha-hectare and the other is km2. The authors should only use the SI measurement system and also km2. If you want to keep ha, it must be done throughout the text!

Due to the very large number of short scientific terms (abbreviations), it is mandatory for authors to add a special section called Abbreviations. This section can be inserted before the References section.

Section Introduction

Between lines 76 and 78 the authors state that they analyzed satellite imagery and NDVI (Normalized Differences Vegetation Index), and in the next sentence they explain the use of MODIS (Moderate Resolution Imaging Spectroradiometer). But it is not clear to me how they used the data for NDVI analysis for MODIS, because first it is moderate resolution and second it is a radiospectrometer? Please explain better.

In lines 88 and 89, the authors need to better explain the chlorophyll absorbance ratio index. My question to them is also how they calculate the health status of the plants. I know that this index varies from -1 to 0 (dead plant or object), 0-0.33 (unhealthy plant), 0.33-0.66 (moderately healthy plant), and 0.66-1.0 (very healthy plant).

Lines 112 and 116, can the authors better explain if they used a DEM (Digital Elevation Model) for slope or not?

Slope is a very important factor for growing crops, especially in high altitude regions.

In general, I think the authors should add more conclusions about Himalayan bioclimate and hydroclimatic factors in this part of the manuscript. The word hydroclimatic factors occurs throughout the text of the manuscript and it is useful to explain more about the hydroclimatic conditions in the Himalayan regions.

Since the authors have used several terms in this section such as NDVI, forests, grasslands, land cover, remote sensing, etc., I strongly recommend reading and citing two valuable references that better explain the methods and techniques of GIS and remote sensing.

These recommended references are

Valjarević, A., Djekić, T., Stevanović, V., Ivanović, R., & Jandziković, B. (2018). GIS numerical and remote sensing analyses of forest changes in the Toplica region for the period of 1953–2013. Applied geography, 92, 131-139. https://doi.org/10.1016/j.apgeog.2018.01.016.

Barmpoutis, P., Papaioannou, P., Dimitropoulos, K., & Grammalidis, N. (2020). A review on early forest fire detection systems using optical remote sensing. Sensors, 20(22), 6442. https://doi.org/10.3390/s20226442.

Section Study Area

Line 164, please explain better the term subtropical area.

Figure 1, for me it is necessary to add geographical coordinates (longitude and latitude) on this map. So, do this.

Also, the authors need to explain how they estimated the position of the glaciers

The authors stated in the text that they used Google Earth Engine, but I think that this map (Figure 1), made with QGIS software, the authors can explain better.

Table 1, can the authors explain from which source they analyzed and divided type of forest land?

Section Hydroclimatic Datasets

In this section, it is not clear whether the authors used data from terrestrial meteorological stations If so, they need to better explain the use of mountain meteorological station data

MODIS daily cloud-free snowpack product section

In my opinion, this section needs to be rewritten. It is not well explained how the authors manipulated with the Landsat8 satellite imagery and then with the data downloaded from the MODIS mission.

The new section needs to be called Statistical or Geostatistical Analysis.

This is because the authors used MK (Mann-Kendall trend test for analyzing trends).

Also in this part of the manuscript I did not see how important the value of (p) is when someone uses the MK trend test?

Fig. 4-6.

I recommend that the authors also present the seasonal results.

I did not find in the manuscript how the authors estimated evapotranspiration? Did they use data from the satellite missions?

Discussion section

In this section, the authors need to add and compare already published similar research. Also, authors can compare the very similar methodology with the very similar research papers.

Can the authors provide some information for other parts of the Himalayas?

Conclusion section

The reply to the authors can be addressed to

Why is this research important for land cover change in the Indian part of the Himalayan Mountains?

Explain better the numerical and remote sensing methodology?

The authors stated in the keywords that they use Google Earth, but I didn’t see in the text that they use it?

Did the authors use this methodology to establish a relationship between climate change impacts and estimated values for forest and grassland increase and loss?

This work has the potential to be published. The authors have done a lot in this manuscript. The work is very interesting and scientifically correct.

In the end, I recommend a major revision

Good luck to the authors

Reviewer#2

Author Response

(The authors gave the same response as above.)

Reviewer 3 Report

Comments and Suggestions for Authors

Comments on the Quality of English Language

Author Response

(The authors gave the same response as above.)

Round 2

Reviewer 1 Report

Comments and Suggestions for Authors

Thank you for improving you work.

Author Response

I am very thankful to the learned reviewer for dedicating their valuable time to review my paper and providing constructive comments to enhance the manuscript's quality. Indeed, the quality of this manuscript has improved significantly. I have done my utmost to incorporate your suggestions into the revised manuscript. Once again, thank you very much to learned reviewer for accepting my manuscript. 

Reviewer 2 Report

Comments and Suggestions for Authors

The manuscript entitled Spatiotemporal Vegetation Variability and Linkage with Snow-Hydroclimatic Factors Using Remote Sensing in Western Himalaya. In my opinion, this manuscript can be accepted in its present form.

Sincerely,  

Reviewer #1

Author Response

(The authors gave the same response as above.)

Reviewer 3 Report

Comments and Suggestions for Authors

The paper “Spatiotemporal vegetation variability and linkage with snow- 2 hydroclimatic factors using remote sensing in Western Himalaya” is an important study because Himalayan ecosystem is of immense importance. The paper is well written but needs some justification

(a)    MODIS is a course resolution dataset how does MODIS justify your study kindly elaborate

(b)   Why other datasets like Landsat, Sentinel time series have not been considered

(c)    Time period 2001 to 2019 is explored why not extended till date to complete two decades

(d)   In Figure 3, 4 and 5 the average line is not clear enough

(e)    Scales are missing in the map

(f)    Add a table to understand the statistical variation in the region

(g)   Produce Heatmaps to represent co relation coefficient

(h)   Refer Following papers for better interpretation and cite them

1.      Agricultural drought conditions over mainland Southeast Asia: Spatiotemporal characteristics revealed from MODIS-based vegetation time-series. International Journal of Applied Earth Observation and Geoinformation, 121, 103378.

2.      Association between drought and agricultural productivity using remote sensing data: a case study of Gujarat state of India. Journal of Water and Climate Change, 11(S1), 189-202.

3.      Large uncertainties in precipitation exert considerable impact on land surface temperature modeling over the Tibetan Plateau. Journal of Geophysical Research: Atmospheres, e2022JD037615.

4.      Spatio-temporal changes in NDVI and rainfall over Western Rajasthan and Gujarat region of India. Journal of Agrometeorology, 20(3), 189-195.

Author Response

I am very thankful to the learned reviewer for dedicating their valuable time to review my paper and providing constructive comments to enhance the manuscript's quality. Indeed, the quality of this manuscript has improved significantly. I have done my utmost to incorporate your suggestions into the revised manuscript. If there are any further recommendations, please do not hesitate to let me know, and I will certainly include them in the final version of this manuscript. Once again, thank you very much.
